# Effects of a Hatchery Byproduct Mixture on Growth Performance and Digestible Energy of Various Hatchery Byproduct Mixtures in Nursery Pigs

**DOI:** 10.3390/ani10010174

**Published:** 2020-01-20

**Authors:** Jung Yeol Sung, Beob Gyun Kim

**Affiliations:** Department of Animal Science and Technology, Konkuk University, Seoul 05029, Korea; jungyeolsung@gmail.com

**Keywords:** energy, growth performance, hatchery byproduct mixture, swine

## Abstract

**Simple Summary:**

There is limited information on the optimal inclusion rate of a hatchery byproduct mixture and digestible energy concentrations in different hatchery byproduct mixtures in nursery pigs. The objectives of this study were to determine effects of a hatchery byproduct mixture on growth performance and to determine digestible energy concentrations of hatchery byproduct mixtures in nursery pigs. Growth performance of pigs fed a hatchery byproduct mixture up to 10% was not different from pigs fed a control diet. Digestible energy concentrations differed among hatchery byproduct mixtures. Based on the current results, the hatchery byproduct mixture can be used in nursery pig diets and different energy values should be applied to different hatchery byproduct mixtures.

**Abstract:**

The objectives were to determine effects of a hatchery byproduct mixture (HBM) on growth performance and to measure digestible energy concentrations in various HBM. In the growth performance experiment, 96 pigs (initial body weight = 9.6 ± 0.8 kg) were assigned to 4 dietary treatments in a randomized complete block design with 6 blocks. Each treatment consisted of 6 replicate pens with 4 pigs comprising 2 barrows and 2 gilts. Pigs were fed graded concentrations of HBM at 0%, 3.33%, 6.67%, and 10.00% for 14 days. In the energy digestibility experiment, 10 barrows (initial body weight = 11.5 ± 0.4 kg) were employed to determine digestible energy in HBM. A basal diet based on corn and soybean meal and 4 additional diets containing 25% of 4 different HBM were prepared. The marker-to-marker method was employed for total collection and the experimental design was a replicated 5 × 4 Latins square design. Growth performance was not compromised as the inclusion rate of HBM increased up to 10%. Digestible energy of HBM ranged from 2772 to 3887 kcal/kg as-is basis. In conclusion, HBM can be used in nursery pig diets and different energy values should be used for each HBM.

## 1. Introduction

Animal protein sources such as fish meal and spray-dried plasma protein are widely used in nursery pig diets to maximize growth performance of pigs. However, as these ingredients are expensive and the price fluctuates, alternative protein sources are required to save swine production costs. Hatchery byproducts, wastes from hatchery facilities, include infertile eggs, unhatched eggs, culled chicks, and eggshells [1]. Considering that hatchery byproducts contain approximately 30% to 60% of crude protein [2], hatchery byproducts are regarded as alternative feed ingredients which can replace widely-used animal protein sources in nursery pig diets [3].

In previous studies, nutritional values of hatchery byproducts including infertile eggs, unhatched eggs, culled chicks, and a mixture of 3 ingredients fed to nursery pigs were determined [2,4]. As hatchery byproducts are discarded together, it is likely that a hatchery byproduct mixture (HBM) is used in swine diets rather than a single hatchery byproduct. However, to our knowledge, information on the effects of HBM on growth performance in nursery pigs is very limited [3], which challenges the swine industry to determine the inclusion rate of HBM in swine diets.

In the previous study, energy concentration of the HBM was determined, which contained 20% of dried infertile eggs, 20% of dried unhatched eggs, and 60% of dried culled chicks to mimic a natural product in a layer facility where hatchery byproduct ingredients were obtained [2]. However, the ratio of ingredients in HBM is variable according to hatchery facility conditions. For precise feed formulations, energy concentrations in feed ingredients should be accurately evaluated [5], but information on available energy in various HBM is very limited.

For these reasons, the objectives of the present study were to determine the effects of HBM on growth performance and to determine digestible energy (DE) concentrations in various HBM in nursery pigs.

## 2. Materials and Methods

All protocols used in the study were approved by the Animal Care and Use Committee of Konkuk University (approval number: KU19054 and KU19058).

### 2.1. Preparation of Hatchery Byproducts

Grinding and drying processes were identical to Sung et al. [2]. Briefly, infertile eggs, unhatched eggs, and culled chicks were obtained from a layer hatchery facility (Join Inc., Pyeongtaek, Republic of Korea). Each ingredient was ground and then dried at 130 °C for 20 h in a dryer (DN 2300, Dongnam Tech Inc., Hwaseong, Republic of Korea). Infertile eggs and unhatched eggs included eggshells. After the drying process, 5 different HBM were prepared by mixing infertile eggs, unhatched eggs, and culled chicks with different ratios (Table 1).

### 2.2. Animals, Diets, and Experimental Design

The growth performance experiment consisted of 2 batches. In batch 1, a total of 48 pigs (24 barrows and 24 gilts) weaned at 28 days of age with initial body weight (BW) of 9.4 kg (standard deviation = 0.7) were used. Pigs were assigned to 4 dietary treatments in a randomized complete block design with 3 blocks based on BW using a spreadsheet program developed by Kim and Lindemann [6]. Each dietary treatment consisted of 3 replicate pens with 4 pigs consisted of 2 barrows and 2 gilts. Four corn-soybean meal-based diets were prepared to contain 0%, 3.33%, 6.67%, and 10.00% of HBM1 to replace fish meal and spray-dried plasma protein in diets (Table 2). Hatchery byproduct mixture1 consisted of 20% of dried infertile eggs, 20% of dried unhatched eggs, and 60% of dried culled chicks to mimic a natural product in a layer facility where the hatchery byproduct ingredients were obtained. In batch 2, forty-eight pigs (9.8 ± 0.9 kg initial BW) weaned at 28 days of age consisted of 24 barrows and 24 gilts were employed. The experimental design and conditions of batch 2 were identical to those of batch 1. Overall, 96 pigs were used and each dietary treatment consisted of 6 replicate pens.

In the energy digestibility experiment, 10 barrows (11.5 ± 0.4 kg initial BW) weaned at 28 days of age were used. A basal diet based on corn and soybean meal was prepared, and 4 additional diets were prepared by mixing 75% of the basal diet with 25% of HBM2, HBM3, HBM4, and HBM5, respectively (Table 3). The ratio of corn, soybean meal, dried whey, and fish meal was the same in all experimental diets. These 5 experimental diets were fed to 10 pigs employing a replicated 5 × 4 incomplete Latin square design with 10 pigs, 5 diets, and 4 periods resulting in 8 replicates per treatment [7].

### 2.3. Feeding, Measurements, and Sample Collection

In the growth performance experiment, each pen was equipped with a feeder and a nipple drinker and pigs had free access to feed and water. On day 7 and 14, individual BW of pigs and feed disappearance in each pen were recorded.

In the energy digestibility experiment, all pigs were individually housed in metabolism crates (0.35 × 1.00 × 1.50 m) equipped with a feeder and a nipple drinker. The quantity of feed provided daily per pig was calculated as 2.7 times the estimated energy requirement for maintenance (i.e., 106 kcal of metabolizable energy per kg BW^0.75^) [8] and divided into 2 equal meals at 08:00 and 17:00 h. Each period consisted of a 5-day adaptation period and a 5-day collection period. During the collection periods, a total collection method was used to collect feces with the marker-to-marker procedure [9]. Chromium oxide and ferric oxide were used as an indigestible marker for the initiation and the termination of fecal collection, respectively. During the collection period, refused feeds were also collected to calculate the amount of actual feed intake.

### 2.4. Chemical Analyses

Dry matter [10], crude protein (method 990.03), ash (method 942.05), calcium (Ca; method 935.13), and phosphorus (method 946.06) in ingredients and diets were determined [11]. Concentrations of gross energy (GE) in ingredients and diets were determined using a bomb calorimetry (Parr 1261, Parr Instruments Co., Moline, IL, USA). In the energy digestibility experiment, feces collected were analyzed for GE.

### 2.5. Calculations and Statistical Analyses

In the growth performance experiment, average daily gain (ADG), average daily feed intake (ADFI), and gain to feed within each treatment were calculated based on mean BW of pigs and consumed diet in each pen. In the energy digestibility experiment, the difference procedure was used to calculate energy concentrations and digestibility of HBM [9]. Predicted energy digestibility of HBM was calculated based on energy digestibility in infertile eggs, unhatched eggs, and culled chicks reported by Sung et al. [2] and ratios of the ingredients in each HBM.
Predicted energy digestibility (%) of HBM2 (50% infertile eggs + 50% unhatched eggs) = (0.5 × DE_infertile eggs_ + 0.5 × DE_unhatched eggs_) ÷ (0.5 × GE_infertile eggs_ + 0.5 × GE_unhatched eggs_) × 100%
where DE_infertile eggs_, DE_unhatched eggs_, GE_infertile eggs_, and GE_unhatched eggs_ are DE and GE values of infertile eggs, and unhatched eggs, respectively, determined by Sung et al. [2]. Predicted energy digestibility of other HBM was determined in the same manner based on ratios of the ingredients in each HBM.

Data were analyzed using the MIXED procedure of SAS (SAS Inst. Inc., Cary, NC, USA). In the growth performance experiment, data from 2 batches of experiments employing the same dietary treatment were pooled for statistical analysis. Dietary treatments were considered as a fixed variable while batch and block within batch were considered as random variables. Orthogonal polynomial contrast was used to analyze linear and quadratic effects of HBM1 on BW, ADG, ADFI, and gain to feed. Least square means for each treatment were calculated and the experimental unit was a pen.

In the energy digestibility experiment, dietary treatments were considered as a fixed variable, while replication, animal within replication, and period within replication were considered as random variables. The least square means were calculated for each treatment, and differences between least squares means were tested using the PDIFF option with Tukey’s adjustment [12]. The experimental unit was a pig, and an alpha level of 0.05 was used to determine statistical significance.

## 3. Results

In the batch 1 in the growth performance experiment, 2 pigs of block 3 in a pen allotted to a control diet died. As only 2 pigs remained in the pen, the pen was excluded from the data. With the inclusion rate of HBM1 up to 10% in diets, no significant differences were observed in final BW, ADG, ADFI, and gain to feed (Table 4).

In the energy digestibility experiment, one pig fed the HBM2 diet had diarrhea during the collection period and this observation was excluded from the data. Diet intake of pigs fed the HBM2 diet was greater (*p* < 0.05) than in pigs fed the HBM5 diet (Table 5). Energy digestibility of basal diet was greater (*p* < 0.05) than that of other experimental diets (*p* < 0.05). Digestible energy of HBM2 diet was the lowest (*p* < 0.05) among experimental diets.

The DE of HBM2 was the lowest and DE of HBM4 was greater than that of HBM3 (*p* < 0.05; Table 6). Determined energy digestibility of HBM4 was greater (*p* < 0.05) compared with HBM2. The determined and predicted energy digestibility values of HBM4 were very close (78.2% vs. 77.9%, respectively).

## 4. Discussion

To overcome weaning stress and maximize growth performance of nursery pigs, animal protein sources such as fish meal and spray-dried plasma protein are routinely included in nursery pig diets [13,14]. As animal protein sources are digested better compared with plant-derived protein sources, nursery pigs would benefit from the efficient provision of essential amino acids provided by animal protein sources. In addition to nutritional aspects, spray-dried plasma protein has been reported to improve immune systems of piglets by providing immunoglobulins and glycoproteins [15,16]. In the present growth performance experiment, HBM1 replaced both fish meal and spray-dried plasma protein in diets without negative effects on growth performance. However, it is likely that HBM may not replace spray-dried plasma protein in nursery pig diets in the industry due to its potentials to enhance immune competence.

Due to safety issues on hatchery byproducts, these ingredients used to be prohibited in animal feeds. Currently, however, hatchery byproducts are allowed in many countries including Republic of Korea, the United States, and the European Union [1,17]. Before conducting the present study, microbial analyses on hatchery byproduct ingredients were performed resulting in no evidence of spoilage [18].

Chemical compositions of hatchery byproduct are highly variable in the literature [2,3,19], which is likely due to variation in the ratios of hatchery byproduct ingredients and processing methods. Gross energy of hatchery byproducts used in the present work were less than that of byproducts used in Sung et al. [2] even though these ingredients were processed using the same method. The reason for the different energy concentrations is unclear.

The present growth performance results are not consistent with Adeniji and Adesiyan [3] who reported negative impacts of HBM on growth performance in nursery pigs. In the study of Adeniji and Adesiyan [3], pigs were fed graded concentrations of HBM (22% crude protein and 32% ash) at 0%, 7.5%, 15.0%, 22.5% and 30.0% for 21 days, and the ADG and ADFI decreased in pigs fed diets containing HBM compared with pigs fed the control diet. Adeniji and Adesiyan [3] attributed the compromised growth performance to unpleasant odor of the HBM used in their study. Moreover, even though the inclusion rate of the hatchery byproduct increased in diets, the inclusion rate of fish meal (4.5%), bone meal (2.5%), and oyster shell (1.0%) were constant resulting in excessive Ca concentration in diets. As excessive dietary Ca over the Ca requirement may negatively affect mineral digestibility and growth performance [20,21], hatchery byproducts should be included in swine diets not to exceed the Ca requirement by limiting inorganic Ca sources such as limestone. In contrast to the study of Adeniji and Adesiyan [3], Ca concentrations in diets in the present study were formulated to be the same among experimental diets. Even though a discrepancy on analyzed Ca concentration in diets existed (0.72% to 0.91% as-fed basis), the impact of this discrepancy might be little considering that no significant differences were observed in growth performance.

As hatchery byproducts are disposed of together, it is likely that available hatchery byproducts from hatchery facilities are mixtures of hatchery byproduct ingredients. In the present study, various HBM were formulated based on different ratios of ingredients to contain a wide range of GE. Diet intake in pigs fed the HBM2 diet was greater compared with pigs fed the HBM5 diet (682 vs. 634 g/d) whereas GE in the HBM2 diet was less than that in the HBM5 diet (3885 vs. 4069 kcal/kg as-fed basis). The different feed intake between HBM2 and HBM5 diets may be attributed to energy concentrations because pigs consume feed until their energy requirement is met [22]. Energy digestibility of experimental diets was not different but DE in diets differed mainly due to differences in GE concentrations in diets.

Even though no difference was observed in energy digestibility of experimental diets, energy digestibility of HBM4 was greater than that of HBM2. This discrepancy is partially explained by the difference procedure which was used for calculating energy digestibility of HBM. Differences between energy digestibility of experimental diets are amplified in differences of test ingredients and the discrepancy becomes greater when the inclusion rate of test ingredients is low [23].

The reason for the lower DE of HBM2 is mainly attributed to lower GE. Hatchery byproduct mixture2 consisted of 50% of dried infertile eggs and 50% of dried unhatched eggs which contained eggshells. As eggshells contain very little energy and the utilization of eggshells is very poor, GE of HBM2 was less than GE of other HBM and fecal output of pigs fed the HBM2 diet was greater than that of pigs fed the HBM5 diet.

Determined and predicted energy digestibility of HBM4 were comparable (78.2% vs. 77.9%) whereas determined values were less than predicted values in HBM2 (67.7% vs. 76.2%), HBM3 (72.8% vs. 78.7%), and HBM5 (75.0% vs. 83.4%). Except for HBM4 containing infertile eggs and culled chicks, other HBM contained unhatched eggs. Considering determined and predicted energy digestibility and the ratio of ingredients in HBM, energy digestibility values of infertile eggs and culled chicks in the current study were comparable to values determined in the study of Sung et al. [2]. In contrast to infertile eggs and culled chicks, energy digestibility of unhatched eggs in the present study were less than the reported value by Sung et al. [2]. This is also supported by numerically greater energy digestibility of HBM4 (33% infertile eggs + 67% culled chicks; 78.2%) over HBM5 (33% unhatched eggs + 67% culled chicks; 75.0%). This numerical difference indicates that energy digestibility in infertile eggs might be greater than in unhatched eggs in the current study. However, in the study of Sung et al. [2], infertile eggs (66%) had lower energy digestibility compared with unhatched eggs (87%). Therefore, energy digestibility of unhatched eggs in the current study may be less than that in the Sung et al. [2], but the reason for this is not clear. The relatively low energy digestibility of unhatched eggs compared with infertile eggs and culled chicks used in the present study likely resulted in lower energy digestibility of HBM2 which contained 50% of unhatched eggs. The relatively low energy digestibility of HBM2 is one the reasons for the low DE in HBM2. Further studies are warranted to identify other reasons for the low DE in HBM2.

## 5. Conclusions

A hatchery byproduct mixture can fully replace fish meal and spray-dried plasma protein in nursery pig diets without negative effects on growth performance. As energy utilization of hatchery byproduct mixtures are variable in pigs, different energy values should be used for each hatchery byproduct mixtures. Further studies are warranted to minimize nutrient variability of hatchery byproducts and to determine effects of hatchery byproducts on the pig performance during the growing-finishing period.

## Figures and Tables

**Table 1 animals-10-00174-t001:** Analyzed composition of hatchery byproduct mixtures, as-is basis.

Item	Hatchery Byproduct Mixture (HBM)				
	HBM1	HBM2	HBM3	HBM4	HBM5
**Composition of HBM, %**					
Dried infertile eggs	20.0	50.0	33.3	33.3	-
Dried unhatched eggs	20.0	50.0	33.3	-	33.3
Dried culled chicks	60.0	-	33.3	66.7	66.7
Total	100.0	100.0	100.0	100.0	100.0
Dry matter, %	94.1	95.6	94.3	93.4	93.3
Gross energy, kcal/kg	4734	4093	4553	4975	4753
Crude protein, %	54.3	37.5	47.1	57.8	57.4
Ash, %	19.5	35.4	25.8	16.7	18.6
Calcium, %	5.6	12.2	8.5	4.9	5.3
Phosphorus, %	0.89	0.58	0.79	1.10	1.11

**Table 2 animals-10-00174-t002:** Ingredients and chemical compositions of the experimental diets, as-fed basis (growth performance experiment).

Item	Inclusion Rate of Hatchery Byproduct Mixture1 ^1^			
	0	3.33	6.67	10.00
Ingredient, %				
Corn	53.53	52.98	52.43	51.79
Soybean meal	15.00	15.00	15.00	15.00
Fermented soybean meal	5.00	5.00	5.00	5.00
Whey powder	15.00	15.00	15.00	15.00
Fish meal	4.00	2.67	1.33	0.00
Spray-dried plasma protein	4.00	2.67	1.33	0.00
Hatchery byproduct mixture1	0.00	3.33	6.67	10.00
l-Lysine-HCl, 78.8%	0.28	0.38	0.47	0.57
dl-Methionine, 99%	0.08	0.08	0.09	0.09
l-Threonine, 99%	0.05	0.08	0.11	0.14
l-Tryptophan, 99%	0.00	0.00	0.01	0.03
l-Valine, 99%	0.00	0.00	0.04	0.09
Soybean oil	1.00	1.00	1.00	1.00
Monosodium phosphate	0.14	0.29	0.40	0.57
Ground limestone	1.27	0.87	0.47	0.07
Sodium chloride	0.40	0.40	0.40	0.40
Vitamin-mineral premix ^2^	0.25	0.25	0.25	0.25
Analyzed composition				
Dry matter, %	90.0	90.3	90.3	90.5
Gross energy, kcal/kg	3982	3997	4019	4056
Crude protein, %	22.7	21.8	22.1	22.3
Ash, %	5.8	5.9	5.9	6.1
Calcium, %	0.72	0.79	0.85	0.91
Phosphorus, %	0.56	0.57	0.59	0.61

^1^ A mixture of 20% dried infertile eggs, 20% dried unhatched eggs, and 60% dried culled chicks. ^2^ Provided the following quantities per kilogram of complete diet: vitamin A, 5000 IU; vitamin D_3_, 1000 IU; vitamin E, 0.6 IU; vitamin K, 0.3 mg; thiamin, 0.3 mg; riboflavin, 0.8 mg; pyridoxine, 0.5 mg; vitamin B_12_, 0.003 mg; pantothenic acid, 2.5 mg; folic acid, 0.5 mg; niacin, 5.0 mg; biotin, 0.05 mg; Cu, 1.3 mg as copper sulfate; Fe, 10 mg as iron sulfate; I, 0.6 mg as calcium iodate; Mn, 30 mg as manganese sulfate; Zn, 38 mg as zinc sulfate; Co, 0.3 mg as cobaltous carbonate; Mg, 5.0 mg as magnesium oxide; and choline chloride, 63 mg.

**Table 3 animals-10-00174-t003:** Ingredients and chemical compositions of the experimental diets containing various hatchery byproduct mixtures (HBMs), as-fed basis (energy digestibility experiment).

Item	Diet ^1^				
	Basal	HBM2	HBM3	HBM4	HBM5
Ingredient, %					
Corn	75.35	56.39	56.39	56.39	56.39
Soybean meal	10.00	7.48	7.48	7.48	7.48
Whey powder	10.00	7.48	7.48	7.48	7.48
Fish meal	4.00	2.99	2.99	2.99	2.99
HBM2	-	25.00	-	-	-
HBM3	-	-	25.00	-	-
HBM4	-	-	-	25.00	-
HBM5	-	-	-	-	25.00
Sodium chloride	0.40	0.40	0.40	0.40	0.40
Vitamin-mineral premix ^2^	0.25	0.25	0.25	0.25	0.25
Analyzed composition					
Dry matter, %	87.9	90.1	89.5	89.5	89.4
Gross energy, kcal/kg	3859	3885	3975	4128	4069
Crude protein, %	14.1	20.5	22.2	24.1	25.3
Ash, %	3.6	12.5	9.3	6.8	7.2
Calcium, %	0.33	3.61	2.63	1.51	1.88
Phosphorus, %	0.41	0.49	0.48	0.53	0.57

^1^ HBM2 = a mixture of 50% dried infertile eggs and 50% dried unhatched eggs; HBM3 = a mixture of 33.3% dried infertile eggs, 33.3% dried unhatched eggs, and 33.3% dried culled chicks; HBM4 = a mixture of 33.3% dried infertile eggs and 66.7% dried culled chicks; HBM5 = a mixture of 33.3% dried unhatched eggs and 66.7% dried culled chicks. ^2^ Provided the following quantities per kilogram of complete diet: vitamin A, 5000 IU; vitamin D_3_, 1000 IU; vitamin E, 0.6 IU; vitamin K, 0.3 mg; thiamin, 0.3 mg; riboflavin, 0.8 mg; pyridoxine, 0.5 mg; vitamin B_12_, 0.003 mg; pantothenic acid, 2.5 mg; folic acid, 0.5 mg; niacin, 5.0 mg; biotin, 0.05 mg; Cu, 1.3 mg as copper sulfate; Fe, 10 mg as iron sulfate; I, 0.6 mg as calcium iodate; Mn, 30 mg as manganese sulfate; Zn, 38 mg as zinc sulfate; Co, 0.3 mg as cobaltous carbonate; Mg, 5.0 mg as magnesium oxide; and choline chloride, 63 mg.

**Table 4 animals-10-00174-t004:** Effects of hatchery byproduct mixture (HBM)1 on growth performance of pigs ^1,2^.

Item	Inclusion Rate of HBM1 ^3^, %				SEM	*p*-Value	
	0	3.33	6.67	10.00		Linear	Quadratic
Day 0 to 7							
Initial BW, kg	9.6	9.5	9.6	9.6	0.3	0.542	0.609
Final BW, kg	11.7	11.7	11.5	11.6	0.4	0.323	0.625
ADG, g/d	310	311	271	286	34	0.225	0.738
ADFI, g/d	481	451	483	430	22	0.182	0.572
Gain to feed	0.64	0.69	0.55	0.66	0.05	0.548	0.209
Day 7 to 14							
Final BW, kg	14.7	14.8	14.4	14.4	0.8	0.419	0.839
ADG, g/d	420	444	416	401	60	0.581	0.557
ADFI, g/d	714	682	671	616	56	0.245	0.839
Gain to feed	0.59	0.66	0.61	0.67	0.07	0.383	0.898
Day 0 to 14							
Initial BW, kg	9.6	9.5	9.6	9.6	0.3	0.542	0.609
Final BW, kg	14.7	14.8	14.4	14.4	0.8	0.419	0.839
ADG, g/d	365	378	343	344	46	0.353	0.781
ADFI, g/d	597	567	577	523	37	0.201	0.747
Gain to feed	0.61	0.67	0.59	0.66	0.06	0.610	0.731

SEM = standard error of the means; BW = body weight; ADG = average daily gain; ADFI = average daily feed intake. ^1^ Experimental unit was the pen with 4 pigs (2 barrows and 2 gilts) per pen. ^2^ Each least squares mean represents 6 observations except for the control diet (5 observations). ^3^ HBM1 = a mixture of 20% dried infertile eggs, 20% dried unhatched eggs, and 60% dried culled chicks.

**Table 5 animals-10-00174-t005:** Energy digestibility of pigs fed experimental diets, as-fed basis ^1,2^.

Item	Diet ^3^					SEM	*p*-Value
	Basal	HBM2	HBM3	HBM4	HBM5		
Diet intake, g/d	675 ^ab^	682 ^a^	660 ^ab^	649 ^ab^	634 ^b^	5	0.029
GE intake, kcal/d	2610	2645	2621	2680	2580	201	0.981
Dry feces output, g/d	56 ^c^	108 ^a^	91 ^ab^	73 ^bc^	76 ^b^	7	<0.001
GE in dry feces, kcal/kg	4595 ^a^	3792 ^c^	4186 ^b^	4766 ^a^	4727 ^a^	86	<0.001
Fecal GE output, kcal/d	255 ^b^	405 ^a^	381 ^a^	346 ^a^	359 ^a^	29	<0.001
Energy digestibility, %	90.1 ^a^	84.7 ^b^	85.7 ^b^	86.7 ^b^	86.0 ^b^	0.8	<0.001
DE in diet, kcal/kg	3478 ^b^	3291 ^c^	3406 ^b^	3578 ^a^	3501 ^ab^	34	<0.001

SEM = standard error of the means; GE = gross energy; DE = digestible energy. ^a–c^ Least squares means within a row without a common superscript differ (*p* < 0.05). ^1^ Each least squares mean represents 8 observations except for HBM2 diet (7 observations). ^2^ Diet intake and fecal output were based on 5 days of collection. ^3^ HBM2 = a mixture of 50% dried infertile eggs and 50% dried unhatched eggs; HBM3 = a mixture of 33.3% dried infertile eggs, 33.3% dried unhatched eggs, and 33.3% dried culled chicks; HBM4 = a mixture of 33.3% dried infertile eggs and 66.7% dried culled chicks; HBM5 = a mixture of 33.3% dried unhatched eggs and 66.7% dried culled chicks.

**Table 6 animals-10-00174-t006:** Energy digestibility of hatchery byproduct mixtures fed to pigs ^1^.

Item	Hatchery Byproduct Mixture (HBM)				SEM	*p*-Value
	HBM2	HBM3	HBM4	HBM5		
**Composition of HBM, %**						
Dried infertile eggs	50.0	33.3	33.3	-		
Dried unhatched eggs	50.0	33.3	-	33.3		
Dried culled chicks	-	33.3	66.7	66.7		
Total	100.0	100.0	100.0	100.0		
As-fed basis						
GE, kcal/kg	4093	4553	4975	4753		
DE, kcal/kg	2772 ^c^	3318 ^b^	3887 ^a^	3568 ^ab^	163	<0.001
Dry matter basis						
GE, kcal/kg	4281	4826	5326	5093		
DE, kcal/kg	2899 ^c^	3517 ^b^	4161 ^a^	3823 ^ab^	173	<0.001
Energy digestibility, %						
Determined	67.7 ^b^	72.8 ^ab^	78.2 ^a^	75.0 ^ab^	3.5	0.018
Predicted ^2^	76.2	78.7	77.9	83.4		

SEM = standard error of the means; GE = gross energy; DE = digestible energy. ^a–c^ Least squares means within a row without a common superscript differ (*p* < 0.05). ^1^ Each least squares mean represents 8 observations except for HBM2 (7 observations). ^2^ Calculated using energy digestibility in infertile eggs (66%), unhatched eggs (87%), and culled chicks (82%) reported by Sung et al. [2] and ratios of the ingredients in hatchery byproduct mixtures.

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
