# Peer review of "Effects of a Hatchery Byproduct Mixture on Growth Performance and Digestible Energy of Various Hatchery Byproduct Mixtures in Nursery Pigs"

_animals, 2020, doi:10.3390/ani10010174_

Round 1

Reviewer 1 Report

This manuscript demonstrated effects of hatchery byproduct mixture on growth performance and energy concentrations of various hatchery byproduct mixtures in nursery pigs. It is very interesting and useful to farmers and feed suppliers in pig production. This is a nice paper that is detailed and easy to read. However, I do have some issues which need to be addressed.

There are some errors in English spelling in the manuscript. Please check it again carefully.

Reason of different composition of hatchery by-product should be explained in M&M, mimic different products from the different plants? Or according to the survey?

L 70 provide weaned time

L 80 and 94 change ‘vitamin mix’ ‘vitamin premix’

L 107 provide size of metabolism crates

L 109 2.7 ‘times energy requirement for maintenance’, but Adeola et al. suggested 4% BW for pig. Do you think which one is better? And what is the difference?

Ref. Adeola, O., 2001. Digestion and balance techniques in pigs. In: Lewis, D.J., Southern,

L.L. (Eds.), Swine Nutrition, 2nd ed. CRC Press, New York, pp. 903-916.

L 128 provide specific equation for calculating DE or ME, and nutrients.

L 193-196 rewrite this sentence, ‘Adesiyan…respectively)’

In Table 5 dry feces output in HBM2 and HBM3 were increased, explain it in discussion.

L22-236 discuss more specific reason for difference DE in HBM, e.g. composition of HBM, undigestible ingredients in HBM and so on.

Question: In general, ME and DE values can get from energy measure exp., compare to DE, ME is more commonly used in pig production, why do not you provide ME in this paper?

Author Response

Response to Reviewer Comments

Manuscript ID: animals-682497

Title: Effects of Hatchery Byproduct Mixture on Growth Performance and Energy Concentrations of Various Hatchery Byproduct Mixtures in Nursery Pigs

Reviewer 1

Comments and Suggestions for Authors

This manuscript demonstrated effects of hatchery byproduct mixture on growth performance and energy concentrations of various hatchery byproduct mixtures in nursery pigs. It is very interesting and useful to farmers and feed suppliers in pig production. This is a nice paper that is detailed and easy to read. However, I do have some issues which need to be addressed.

Response: Thank you very much for your time to review the present manuscript. Your encouraging comments and the valuable suggestions on the manuscript are highly appreciated. We have revised our manuscript according to your comments. The specific responses are listed below.

Genereal comments

There are some errors in English spelling in the manuscript. Please check it again carefully.

Response: We have corrected errors in English in the manuscript

Additional changes:

L 2: Inserted “a” to make “a Hatchery Byproduct Mixture”

L 3: Changed “Energy Concentrations” to “Digestible Energy”

L 11: Inserted “a” to make “a hatchery byproduct mixture”

L 13: Inserted “a”

L 15: Inserted “a”

L 20: Inserted “a”

L 22 and throughout the manuscript: trial >> experiment

L 25 and throughout the manuscript: energy measurement trial >> energy digestibility experiment

L 41: Deleted “potential”

L 46: Deleted “ingredient”

L 52: Hatchery facilities conditions >> Hatchery facility conditions

L 63: Changed to “… processes were …”

L 71: 2 experiments >> 2 batches; experiment 1 >> batch 1

L 75: comprising >> consisted of

L 80: experiment 2 >> batch 2

L 81: comprising >> consisted of

L 81: An >> The

L 82: experiment 2 >> batch 2

L 82: experiment 1 >> batch 1

L 111-112: consumed diet >> feed disappearance

L 117: Inserted “the”

L 117: Inserted “the”

L 146: Inserted “batches of”

L 148: experiment and block within experiment >> batch and block within batch

L 149: responses >> effects

L 155: the pig >> a pig

L 157: experiment >> the batch

L 158: As >> With

L 158: Deleted “increased”

L 161: Removed a space following “(HMB)”

L 161: Deleted “(growth performance trial)” as this appears to be redundant

L 171: Deleted “(energy measurement trial)” as this appears to be redundant

L 183: Deleted “(energy measurement trial)” as this appears to be redundant

L 199: In the growth performance experiment in the current study >> In the present growth performance experiment

L 213: Changed to “The present growth performance results are …”

L 234: Rewritten to “The different feed intake between HBM2 and HBM5 diets may be attributed to energy concentrations …”

L 253: might be >> were

L 255: might be >> were

Reason of different composition of hatchery by-product should be explained in M&M, mimic different products from the different plants? Or according to the survey?

Response: Occurrence rates of hatchery by-products vary depending on the hatchery facilities. In the growth performance trial, the hatchery byproduct mixture (HBM1) was prepared considering the natural occurrence rate of these ingredients in the hatchery facility and moisture contents (20% dried infertile eggs, 20% dried unhatched eggs, and 60% dried culled chicks). This information has been added in M&M (L 77-80). In the energy measurement trial, 4 different hatchery byproduct mixtures (HBM2, HMB3, HBM4, and HBM5) were prepared to contain different gross energy concentrations (L 230-232).

Specific comments

L 70 provide weaned time

Response: Provided weaned time (L 72, 80, and 92).

L 80 and 94 change ‘vitamin mix’ ‘vitamin premix’

Response: Changed as suggested (Tables 2 and 3).

L 107 provide size of metabolism crates

Response: Provided the size of 0.35 × 1.00 × 1.50 m (L 114).

L 109 2.7 ‘times energy requirement for maintenance’, but Adeola et al. suggested 4% BW for pig. Do you think which one is better? And what is the difference? Ref. Adeola, O., 2001. Digestion and balance techniques in pigs. In: Lewis, D.J., Southern, L.L. (Eds.), Swine Nutrition, 2nd ed. CRC Press, New York, pp. 903-916.

Response: We trust that ‘x energy requirement for maintenance’ is better because this feed allowance is close to 90% feed intake at ad libitum. If 4% of BW is provided to 10-kg pigs, the feed intake (ME = 3,300 kcal/kg) would be less than 70% ad libitum. If 4% of BW is provided to 100-kg pigs, the animals would consume only approximately 3% of BW. Dr. Adeola also used 3 to 5% of BW for feed allowance depending on the BW range.

L 128 provide specific equation for calculating DE or ME, and nutrients.

Response: Provided the equation as suggested (L 138-144).

L 193-196 rewrite this sentence, ‘Adesiyan…respectively)’

Response: Refined this sentence (L 214-217)

In Table 5 dry feces output in HBM2 and HBM3 were increased, explain it in discussion.

Response: Explained a potential reason in the discussion section (L 243-247).

L 222-236 discuss more specific reason for difference DE in HBM, e.g. composition of HBM, undigestible ingredients in HBM and so on.

Response: Explained as suggested (L 243-247 and 262-266).

Question: In general, ME and DE values can get from energy measure exp., compare to DE, ME is more commonly used in pig production, why do not you provide ME in this paper?

Response: Your comment is correct. ME is more frequently used in the swine industry compared with DE. It is regretful that we do not report ME values. If necessary, hopefully, readers can convert DE to ME using prediction equations available in the literature.

Reviewer 2 Report

The manuscript investivated the effects of hatchery by-product mixture (HBM) inclusion in pig diets on growth performance and energy digestibility. In general, the research design is appropriate, the methods are adequately described, and results are clearly presented. However, some minor revision are recomanded.

The aim of the study should be consistent throughout the MS. i.e. in abstract, determination of energy digestibility is mentioned, however this aspect is lacking in aim description (line-54-55). Accordingly, this part must be implemented.

In the abstract it is not clear the experimental design and dietary treatment. Please improve this part.

In the introduction author did not mentioned safety issue related to hatchery by product and processing technology. a short focus on this topic may be appreciated by the readers.

Also legal status of this product is not mentioned. i.e. in European union it is not allowed as animal feed. Please mention the legal status at least in the country where the study has been conducted. If even in the country where the study has been conducted this material is not allowed the rationality of the study should be revised.

Conclusion section is very short and concise. Author may improve it giving a short remind of the background of hatchery by product and the aim of the study. Furthermore, also the main issue/pitfall related to this product such as nutrient composition variability and safety issue which are not mentioned et all throughout the MS.

Author Response

Reviewer 2

Comments and Suggestions for Authors

The manuscript investivated the effects of hatchery by-product mixture (HBM) inclusion in pig diets on growth performance and energy digestibility. In general, the research design is appropriate, the methods are adequately described, and results are clearly presented. However, some minor revision are recomanded.

Response: Thank you very much for the encouraging comments. Your valuable suggestions are highly appreciated. We have revised our manuscript according to your helpful comments. The specific responses are listed below.

Genereal comments

The aim of the study should be consistent throughout the MS. i.e. in abstract, determination of energy digestibility is mentioned, however this aspect is lacking in aim description (line-54-55). Accordingly, this part must be implemented.

Response: We have changed “energy concentrations” to “digestible energy concentrations” to be consistent with the abstract (L 56 and throughout the manuscript).

In the abstract it is not clear the experimental design and dietary treatment. Please improve this part.

Response: Thank you for this comment. For the growth performance experiment, the experimental design and dietary treatments were clarified. For the energy digestibility experiment, unfortunately, we were unable to explain 4 additional diets in detail due to the word limit (200) of this journal.

In the introduction author did not mentioned safety issue related to hatchery by product and processing technology. a short focus on this topic may be appreciated by the readers.

Response: Provided as suggested (L 205-207).

Also legal status of this product is not mentioned. i.e. in European union it is not allowed as animal feed. Please mention the legal status at least in the country where the study has been conducted. If even in the country where the study has been conducted this material is not allowed the rationality of the study should be revised.

Response: Provided as suggested (L 203-205).

Conclusion section is very short and concise. Author may improve it giving a short remind of the background of hatchery by product and the aim of the study. Furthermore, also the main issue/pitfall related to this product such as nutrient composition variability and safety issue which are not mentioned et all throughout the MS.

Response: We have provided nutrient composition variability (L 208-212) and safety issue (L 203-207). We also have provided the need for further studies in the Conclusion section (L 271-273).

Reviewer 3 Report

Well designed and well presented. However, the length of the experimental feeding period was short.  There is a reasonable explanation why the results were not consistent with Adeniji and Adesiyan, it would be interesting to see if the impacts of The different combinations of hatchery byproduct hold over a longer period of time, and if those nutritional effects hold throughout the time to market weight. 

This is a good paper that will add to the knowledge base needed to utilize byproducts.  More work like this on all available byproducts is needed to enhance sustainability. 

Author Response

Reviewer 3

Comments and Suggestions for Authors

Well designed and well presented. However, the length of the experimental feeding period was short.  There is a reasonable explanation why the results were not consistent with Adeniji and Adesiyan, it would be interesting to see if the impacts of The different combinations of hatchery byproduct hold over a longer period of time, and if those nutritional effects hold throughout the time to market weight.

Response: Thank you for your comments. We absolutely agree that it will be very interesting to see the long-term effects of feeding hatchery byproducts. The short experimental period is a limitation of the performance trial. In the conclusion section we additionally mention that further studies are warranted to trace effects of hatchery byproduct mixtures on the growth performance of pigs until pigs reach their market weight (L 271-273).

This is a good paper that will add to the knowledge base needed to utilize byproducts.  More work like this on all available byproducts is needed to enhance sustainability.

Response: Thank you for the encouraging comment. We agree that more research on available byproduct ingredients is critical for sustainable swine production.